# Characterization, Molecular Mechanism of Prochloraz-Resistance in *Fusarium fujikuroi* and Development of Loop-Mediated Isothermal Amplification Rapid Detection Technique Based on the S312T Genotype of Resistances

**DOI:** 10.3390/jof10080560

**Published:** 2024-08-08

**Authors:** Chenyang Ge, Daixing Dong, Chengxing Mao, Qianqian Zhang, Chuanqing Zhang

**Affiliations:** 1College of Advanced Agricultural Sciences, Zhejiang Agriculture and Forest University, Hangzhou 311300, China; 19857137232@163.com (C.G.); zafumaocx@163.com (C.M.); 2Extension Centre of Agriculture Technology of Fuyang, Hangzhou 311400, China; davidhz2004@163.com; 3Research Institute for the Agriculture Science of Tongxiang, Jiaxing 314500, China

**Keywords:** *Fusarium fujikuroi*, rice bakanae disease, prochloraz resistance, S312T mutation, LAMP

## Abstract

Rice bakanae disease (RBD) is a typical seed-borne fungal disease caused by *Fusarium fujikuroi*. Prochloraz is a sterol demethylation inhibitor, which is among the most important classes of active ingredients for the management of RBD. In 2022, the total resistance frequency of *F. fujikuroi* to prochloraz in Zhejiang Province was 62.67%. The fitness of the prochloraz-resistant population was lower than that of the susceptible population, but its pathogenicity was slightly stronger. The S312T and F511S double mutations of *Ffcyp51b* were detected in the resistant isolates. Loop-mediated isothermal amplification (LAMP) technology based on S312T was established to rapidly determine prochloraz resistance in *F. fujikuroi*. LAMP primer mismatch design was performed based on the *cyp51b* gene, and 100–300 bp sequences containing a mutation at codon 312 were amplified. In a 25 µL reaction tube, 1 pg/µL DNA of *F. fujikuroi* could be detected. The detection limit for the frequency of prochloraz resistance was 0.498% using this method. We performed LAMP detection on rice seedlings inoculated with prochloraz-sensitive and -resistant isolates and treated them with prochloraz. Prochloraz demonstrated good control in rice seedlings. A chromogenic reaction was observed in seedlings treated with prochloraz-resistant isolates, and the results were verified using electrophoresis. It has been demonstrated that LAMP technology based on the S312T genotype can quickly and specifically detect prochloraz-resistant isolates in rice seedlings.

## 1. Introduction

*Fusarium fujikuroi* causes rice bakanae disease (RBD) [1,2], but it can also infect other plants, including maize, soybean, and *Lasia spinosa* [3,4,5]. Among them, RBD is a typical seed-borne fungal disease that occurs in all major rice-producing areas [6]. It damages plants from the seedling stage to the heading stage, and mild occurrence in rice can cause yield losses between 5–50%. Seeds carrying the pathogenic fungus are the main source of RBD infection. When seeds germinate, *F. fujikuroi* can invade seedlings through the bud [7]. With the growth of fungal-bearing seedlings [8], mycelia gradually spread throughout the whole plant, and symptoms, such as internode elongation, weakened tillering ability, yellow leaf color, and abnormal root development, may occur [9,10]. RBD may also occur at the heading stage, and in severe cases, the ears turn brown and fail to set. The use of chemical agents to control RBD through seed treatment [11] is quick, economical, and effective and has been the most important technology for its management.

Prochloraz is a sterol demethylation inhibitor (DMI) [12] that was introduced in China at the end of the 20th century to replace carbendazim, for which fungi had developed serious resistance [13]. Thus far, prochloraz is still the main agent used to control RBD. In 1970, DMI compounds began to be applied to control agricultural diseases, and resistance to DMI fungicides in the field has been reported [14]. Wada observed a decrease in the field efficacy of prochloraz in Japan and reported for the first time that *F. fujikuroi* developed resistance to prochloraz in 1990 [15]. In 2002, Liu analyzed the sensitivity of *F. fujikuroi* isolated from Jiangsu Province to prochloraz. The EC_50_ of six isolates was 10-times higher than that of sensitive isolates (0.005 μg/mL was used as the standard for sensitive isolates) [16]. According to previous studies, the resistance mechanism of *Fusarium* spp. and other pathogens to DMIs includes the *cyp51* gene point mutation, *cyp51* gene overexpression, and the involvement of efflux actions by different transporters [17,18,19]. Zhang showed that the S312T point mutation of *Ffcyp51b* was the main mechanism by which *F. fujikuroi* resists prochloraz [20]. Gao’s experiment also demonstrated this mutation [21]. Li found that two point mutations, S312T and F511S, in *Ffcyp51b* had a synergistic effect on prochloraz resistance in *F. fujikuroi* from Japan [22]. However, this double mutation has not been reported in China. Therefore, resistance assessment is important for RBD control.

Traditional fungicide resistance detection methods are mainly based on the isolation and culture of pathogenic fungi on fungicide-amended plates and identification of resistant isolates according to the inhibitory effects on growth or spore germination [23]. This method requires a long test cycle and many human resources. As an important alternative, polymerase chain reaction (PCR) has been developed to detect fungicide-resistant isolate genotypes. However, cPCR, mPCR, and qRT-PCR detection methods require specially equipped laboratories (equipped with thermal cyclers and real-time PCR machines) and specialized technical expertise [24]. Loop-mediated isothermal amplification (LAMP) [25] is a new nucleic acid amplification technique developed in Japan. LAMP may become a novel alternative to PCR because of its rapidness, high specificity, and low cost [26,27]. It has very broad application prospects. LAMP has been used to detect fungicide resistance in plant pathogenic fungi, including the F200Y mutation for carbendazim-resistant *Sclerotinia sclerotiorum* [28], the E198A mutant genotype [27,29,30], and the *cytb* gene G143A mutation in *Colletotrichum acutatum* and *C. gloeosporioides* [31]. In this study, we seek to understand the evolution of resistance to prochloraz in China and clarify the new mutation genotype associated with prochloraz-resistant in *F. fujikuroi* since 2018. Meanwhile, rapid detection using LAMP was further established based on the S312T point mutation in *cyp51b* to provide technical support for resistance detection, management and the accurate and efficient control of RBD.

## 2. Materials and Methods

### 2.1. Resistance Monitoring and Sensitivity Determination of Fusarium fujikuroi to Prochloraz

The *F. fujikuroi* isolates used in this study were collected from Hangzhou, Jinhua, Ningbo, Shaoxing, and other cities in Zhejiang Province in 2018 and 2022. Potato dextrose agar medium (200 g potatoes, 20 g glucose, 15 g agar, and deionized water up to 1 L, PDA) was used for separation, preservation, and the fungicide sensitivity assay. Prochloraz (99.5%) was obtained from Tianfeng Biotech Corporation (Jinhua, China) [20].

In order to determine the resistance frequency of 75 strains of *Fusarium fujikuroi* to prochloraz, we used the differential dose method. Prochloraz is dissolved in methanol as stock solution firstly and then added to the melted PDA. Fresh plugs (5 mm in diameter) were cut from the growing edge of a mycelial culture and placed on PDA plates (90 mm in diameter) amended with 1 and 5 μg/mL prochloraz, at which concentrations the mycelial growth of the sensitive isolates was completely inhibited, while the resistant isolates continued to grow [21]. The sensitivity to prochloraz of each isolate was determined by the mycelial growth assay. The fresh plugs were placed on PDA plates amended with 1.6, 0.4, 0.1, 0.025, and 0.00625 μg/mL prochloraz. The diameter of each colony was measured perpendicularly after 7 days of incubation at 25 °C, and the median effective concentration (EC_50_) value for each isolate was calculated by linear regression of the inhibition percent of growth relative to the control versus the log10 transformation for each concentration [32,33].

### 2.2. Characterization of Prochloraz-Resistant Isolates and Prochloraz-Sensitive Isolates

#### 2.2.1. Mycelial Growth Rate

The test isolates were cultured on the blank PDA medium in the dark at 25 °C for 5 days. Mycelial plugs (5 mm in diameter) were taken from the edge of the colony, transferred to the non-inoculated PDA medium, and cultured in the dark at 25 °C for 7 days. The colony diameter was measured, and the mycelial growth rate was calculated. Each isolate was repeated three times.

#### 2.2.2. In Vitro Conidial Production and Germination

The resistant and sensitive isolates were used to inoculate PDA plates [23]. After 5 days of culture at 25 °C, mycelial plugs (5 mm in diameter) were taken from the edge of the colony, placed in 100 mL of 3% mung bean soup medium, and shaken at 175 rpm for 3 days at 25 °C. The spores were collected, and the number of conidia was observed by microscopic observation with a hemacytometer. Conidial germination was assessed by plating 100 μL of a conidial suspension (1 × 10^6^ conidia/mL) on 1.5% water agar plates and incubating at 25 °C. After 4, 8, and 12 h, 100 spores were observed under a microscope. The number of germinated spores was counted, and the spore germination rate was calculated. Each isolate comprised three technical replicates, and the experiment was performed three times.

#### 2.2.3. Pathogenicity on Rice Seedlings

Rice seeds were soaked in 70% alcohol for 1 min and in 3% NaClO for 3 min and then rinsed three times with sterile water. The conidial suspension was prepared as described above and cultured at 25 °C for 2–3 d. The germinated seeds were placed in 10 mL of conidial solution (1 × 10^6^ conidia/mL) and shaken at 90 rpm for 12 h at 25 °C. The same treated seeds were placed in sterile water as the control. The seeds were then transplanted into the nutrient solution and cultured at 28 °C under a 12-h light and 12-h dark cycle. Seedling height was measured after 15 days. Each isolate contained 50 rice seedlings, and the overgrowth rate of the rice seedlings was calculated as follows [20]:Overgrowth rate = (Height of treated − Height of control)/Height of control × 100%.

### 2.3. Cloning and Sequencing of Ffcyp51b in Resistant and Sensitive Isolates

Based on the gene sequence of *Ffcyp51b* (FFUJ_01179), primers (BF: 5′ AGGTGTGTGGGGTCTCTCTCT-3′; BR: 5′-AAGCAGCACAGTCGTCATGG-3′) were designed to amplify the 1.80-kbp DNA fragment of the prochloraz-resistant and -sensitive *F. fujikuroi* isolates. The total DNA of each isolate was extracted using a fungal genomic DNA Rapid Extraction Kit (B518229-0100, Sangon Biotech, Shanghai, China), and the DNA concentration was determined using BioDrop μLite (BioDrop, Cambridge, UK). All DNA were stored at −20 °C at Zhejiang A&F University until use to avoid repeated freezing and thawing. The PCR products were directly sequenced, and the results were aligned using Bioedit 7.2.5 software.

### 2.4. Primer Design

The point mutation sequence of the *Cyp51b* gene of *F. fujikuroi* JH1 was used as a template, and a mismatched base was introduced at position 312 using online primer design software Primer Explore V5 [34]. LAMP primers with two external primers (F3 and B3) and two internal primers (FIP and BIP) were designed. The optimal primers were selected according to the ΔG values at the 3′ terminal of F3/B3 and F2/B2, and the ΔG values at the 5′ terminal of F1c and B1c were less than −4 Kcal/mol [35].

### 2.5. PCR Amplification and Sequencing

The 25 μL LAMP reaction system (Table 1) consisted of 0.5 μL Bst DNA (8 U·μL^−1^), 2.5 μL 10× Thermo Pol, 4 μL Mg^2+^ (25 mM), 2.5 μL dNTP (10 mM), 2 μL FIP/BIP (20 μM), 0.75 μL F3/B3 (10 μM), 3 μL betine (5 M), 1 μL HNB (3.75 mM), 1 μL sample DNA, and 5 μL ddH_2_O. LAMP amplification was performed at 63 °C for 90 min and stored at 4 °C [36].

### 2.6. Detection of LAMP Specificity

To investigate the specificity of the LAMP reaction, genomic DNA from fungal isolates or mutants was used (Table 2). After 90 min, the reaction was verified using gel electrophoresis with 1% agarose.

### 2.7. Detection of LAMP Sensitivity

The genomic DNA of *F. fujikuroi* resistant isolate JH-1 was diluted into seven concentrations in a 10-fold gradient: 0.0001, 0.001, 0.01, 0.1, 1, 10, and 100 ng/μL. LAMP amplification was performed using 1 μL DNA template and ddH_2_O as a negative control. The experiment was repeated three times.

### 2.8. Detection Limit of Resistance Frequency in Different Sensitive: Resistant Isolate Ratios

JS16 and JH1 were selected and cultured in conditional medium (CM) at 26 °C in the dark for 5 days. Spore suspensions were prepared and quantified to 10^6^ mL^−1^. DNA was extracted, and the volume ratios of DNA of the two isolates were as follows for JS16:JH1: 10:1, 20:1, 100:1, 200:1, and 400:1. The LAMP test was performed. After 90 min of reaction, 1% agarose was used for gel electrophoresis verification.

### 2.9. Detection of Resistant Isolates from Rice Seeds and Seedlings

To evaluate the feasibility of LAMP technology in detecting the mutant genotype of resistant isolate S312T, resistant isolate JH1 and sensitive isolate JS16 were used to inoculate sterilized seeds. Before inoculation, the two isolates were used to inoculate seeds impregnated with 10 μg/mL prochloraz, sterilized in a water bath at 60 °C for 15 min, and pre-germinated at 30 °C for 48 h. Rice seeds with uniform endosperms were placed in a 20 mL conidia suspension (10^5^ mL^−1^) at 25 °C and shaken at 80 rpm for 20 h. The same treated seeds were placed in sterile water as the control. The seeds were then transplanted into nutrient solution and cultured at 25 °C and 75% humidity in a light incubator (12 h light: 12 h dark) for 15 days to determine the seedling height. Each isolate was replicated in 20 rice seedlings. The average height of the 20 rice seedlings was calculated, and the growth rate of diseased plants was calculated. Three pathogenicity tests were performed on each isolate. After inoculation, two seeds were selected from each treatment for DNA extraction. After the seeds were transplanted and grown into seedlings, two seedlings were selected from each treatment to extract DNA by cutting their internodes. The DNA was detected by LAMP, together with the previous DNA. After 90 min of reaction, gel electrophoresis was performed with 1% agarose.

## 3. Results

### 3.1. Resistance of F. fujikuroi to Prochloraz

Among the 75 tested isolates, 17 had high resistance and 30 had low resistance. The resistance frequency was 22.67 and 40%, respectively, and the total resistance frequency was 62.67% (Figure 1). In 2022, the resistance frequencies of Hangzhou, Shaoxing, Jinhua, Taizhou, and Ningbo were 50, 28.6, 64.3, 66.7, and 87.5%, respectively. Five resistant isolates (F1, F4, F54, F68, and F72), five sensitive isolates (F5, F8, F9, F10, and F25), and nine bakanae isolates collected in 2018 were selected to determine the EC_50_ (Table 2).

### 3.2. Characterization of Prochloraz-Resistant Isolates

Four resistant isolates (F1, F4, F68, and F72) and four sensitive isolates (F9, F10, F25, and F59) were randomly selected to determine the fitness of *F. fujikuroi* to prochloraz. There was a significant difference in the mycelial growth rate between sensitive and resistant isolates. The average mycelial growth rate of the sensitive isolates was 10.19 mm/d, which was higher than that of the resistant isolate (8.27 mm/d). In terms of sporulation, the average sporulation of prochloraz-resistant isolates was 6.74 × 10^6^ spores/mL, which was significantly lower than that of sensitive isolates. The average spore germination rate of resistant isolates at 4, 8, and 12 h (5.22, 30.50, and 80.75%, respectively) was significantly lower than that of sensitive isolates (9.78, 44.50, and 90.89%, respectively). However, resistant isolates were significantly more aggressive than sensitive isolates (Table 3 and Table 4).

### 3.3. Sequence Analysis of Ffcyp51 in F. fujikuroi

The *Ffcyp51b* gene of prochloraz-resistant and -sensitive *F. fujikuroi* isolates was amplified and sequenced (Figure 2). The 511th amino acid F (TTC) was replaced by S (TCC) between Pro-S and Pro-R. In addition to F511S, a previously reported S (TCT) to T (ACT) mutation at amino acid 312 in the prochloraz-resistant isolate was detected at FfCYP51B in Pro-R.

### 3.4. Primer Design

The *cyp51b* (GenBank accession: CP023101.1) containing point mutation sequence of prochloraz wild mutant JH1 was used as a template. Using the online software Primer Explore V5 (https://primerexplorer.jp/e/v5_manual/, accessed on 15 December 2019), a set of specific amplification detection primers (Table 5 and Figure 3), including the outer primer F3/B3 and the inner primer FIP/BIP, were screened according to the LAMP primer design principle. The primers were synthesized by GenScript Corporation, centrifuged at 8000 rpm for 15 s, dissolved in sterile water, and stored at 4 °C.

### 3.5. LAMP Specificity

Chromogenic reactions occurred in both resistant wild-type and site-specific mutant isolates, but not for sensitive isolates and *U. virens*, an important pathogen on rice. The results were consistent with those of electrophoresis (Figure 4a,b), indicating that the established LAMP system had good specificity in detecting prochloraz-resistant isolates (*Cyp51b* gene type 312 codon TCT). For isolates of 2022, chromogenic reactions also only occurred for resistant isolates with S312T mutation in FfCYP51B (Appendix A).

### 3.6. LAMP Sensitivity

The genomic DNA of resistant isolate JH1 in gradient dilution was detected, and the results are shown in Figure 5. Bands were amplified, and a chromogenic reaction occurred when the DNA concentration was higher than 0.001 ng/μL. The results indicated that the detection limit of this LAMP technology was 0.001 ng/μL.

### 3.7. Detection Limit of Resistance Frequency

Upon observing our blank control, sensitive isolate control and when S (FFJS-16S):R (FFJH-1) R) = 400:1, there was no positive reaction. All reactions occurred in R (JH1) (S:R = 10:1; S:R = 20:1; S:R = 100:1; S:R = 200:1), and gel electrophoresis verified that the reaction was not a false positive (Figure 6); that is, the detection limit of resistance frequency by LAMP was 0.498%.

### 3.8. Detection of Prochloraz-Resistant Isolates from Seeds and Seedlings

In this study, the sensitive and resistant isolates were used to inoculate rice seeds at the same time for LAMP detection(Figure 7). The seeds inoculated with sensitive isolates showed a negative reaction, while the seeds inoculated with resistant isolates showed a positive reaction. These results were further confirmed using electrophoresis [37].

Rice seedlings inoculated with both sensitive and resistant isolates showed spindling. There was no significant difference in pathogenicity between the sensitive and resistant isolates. Rice seedlings treated with prochloraz after inoculation with sensitive isolates were consistent with the control, while rice seedlings inoculated with resistant isolates still showed spindling. Prochloraz had a good control effect on malignant seedling disease in rice, but the resistance isolate had no obvious effect. The LAMP test showed positive results for seedlings inoculated with resistant isolates. Therefore, this LAMP system can detect resistant isolates in rice seeds and seedlings.

## 4. Discussion

*Fusarium fujikuroi* is the main pathogen causing RBD. In China, prochloraz has been widely used as a seed treatment fungicide to control RBD for several years, and the frequency of prochloraz resistance has gradually increased. Several studies have reported the resistance of *F. fujikuroi* to prochloraz and its related resistance mechanism [20,38,39]. In the current study, 75 isolates of bakanae pathogens were collected from Zhejiang Province, and their resistance to prochloraz was determined using the discriminatory dose method. The total resistance frequency was 62.67%, including 30 high-resistance isolates and 17 low-resistance isolates. Previous studies have shown that pathogen resistance to fungicides is always accompanied by fitness costs [40,41]. In this study, prochloraz-sensitive isolates were stronger than the resistant isolates in mycelial growth, sporulation, and spore germination, but the pathogenicity was slightly weaker, which may be related to the ability of the isolates to produce toxins.

Findings from a previous study indicated that the resistance mechanism of *F. fujikuroi* to prochloraz is due to the S312T point mutation of *Ffcyp51b* and the overexpression of *Ffcyp51a* and *Ffcyp51b*. In recent years, an amino acid mutation at the 511th position of the *Ffcyp51b* gene sequence has been demonstrated, and the S312T and F511S mutations on *Ffcyp51b* have a synergistic effect on prochloraz resistance of pathogens causing bakanae disease [22]. We amplified the *cyp51b* sequence of prochloraz-resistant isolates isolated in Zhejiang Province in 2022 and found that it was consistent with that found in Japan. There were double mutations of S312T and F511S at the same time, but the sensitivity of our prochloraz-resistant isolates was 10–20 times higher than that previously found. Therefore, the relationship between this newly discovered amino acid site mutation and the resistance of *F. fujikuroi* to prochloraz remains to be further studied.

LAMP is a novel nucleic acid amplification method with high specificity, sensitivity, and simple operation. At present, LAMP has been successfully used to detect bacterial, viral [42,43], and fungal pathogens [44]. The DNA intercalation dye was HNB, and the color change from purple to sky blue indicated a positive reaction, while purple indicated a negative reaction. The reaction times and temperatures were consistent with those in previous studies. By adding HNB (0.15 μM) to the LAMP test mixture, the test results can be displayed. The reaction mixture change from purple to sky blue can be seen with the naked eye, and the test results can be directly and quickly judged without amplification. In addition, the addition of HNB dye did not reduce the sensitivity or specificity of the reaction system.

RBD is a monocyclic disease, which is mainly transmitted through germ-bearing seeds, and seeds with fungi are the main source of initial infection. The main stage of infection is the process of seed soaking and bud promotion. Previous studies on the isolation and identification of Asian rice seeds showed that the fungus-carrying rate ranged from 3 to 92%, and there were great differences among regions. The pathogenic fungi attach to the seed surface in the form of conidia or latent mycelium to overwinter [45]. Therefore, it is of great significance to detect pathogenic fungi at the seed stage. Through the established visual detection methods, seed health can be monitored quickly and accurately, and the seed health status can be effectively evaluated to minimize the threat of pathogenic fungi.

Chemical treatment is the primary method for preventing and controlling RBD [46]. Failure to treat seeds with chemical agents can result in contamination of healthy seeds when soaked with infected ones. The pathogen can then invade rice seedlings through new buds and gradually spread to the whole plant as it grows. Prochloraz is a DMI, which has an obvious control effect on diseases caused by ascomycetes and ascocarps in many crops [47]. Since its introduction in China at the end of the 20th century, prochloraz has become the main chemical control for RBD. Due to the long-term use of a single agent and low concentration of seeds, the pathogen is prone to develop resistance. Studies have shown that resistance to prochloraz and phenamacril has developed to different degrees. If rice seeds with a prochloraz-resistant isolate are treated with prochloraz, RBD would not be effectively controlled.

The traditional detection methods of seed fungi include the water agar method and the isolation and culture method [48]. These detection methods are time-consuming and can only be identified through morphological and biological characteristics combined with the relevant literature, easily resulting in misidentification of similar species. Modern molecular biology technologies include common PCR detection based on target gene sequences and real-time fluorescence quantitative PCR. However, the common characteristics of these technologies are a long cycle time, expensive instruments, and the inability to meet the needs of grassroot detection [49].

The point mutation in the 312th codon of the *cyp51b* gene results in the alteration of the S312T amino acid in *F. fujikuroi*, resulting in high prochloraz resistance. In this study, we developed a LAMP assay to detect the genotype of the prochloraz-resistant S312T mutant of *F. fujikuroi*. Before seed treatment, the seeds were tested, providing technical support for the detection of the S312T genotype of the prochloraz-resistant *F. fujikuroi* mutant before sowing. Based on our results, this LAMP detection method can accurately detect rice seeds and seedlings simulated with different sensitive isolates, providing technical support for the next step in detecting whether rice seeds contain prochloraz-resistant fungi before sowing. For further study, the S312T-LAMP test should be used to collect RBD samples from different regions to avoid the inefficiency of fungicides due to drug resistance. Evolutionary patterns of genotypes in resistant isolates should also be studied to develop new control strategies using combinations of different fungicides to control RBD.

## 5. Conclusions

In this study, we found that the S312T + F511S double mutations are responsible for prochloraz resistance in *F. fujikuroi* after previous report of S312T single mutation. Based on this resistance molecular mechanism, a LAMP system was established to detect the mutation site (*Cyp51b* 312 codon TCT-ACT) in *F. fujikuroi* for prochloraz resistance. This method can rapidly detect prochloraz resistance on rice seeds and seedlings to provide support for the scientific control of RBD.

## Figures and Tables

**Figure 1 jof-10-00560-f001:**
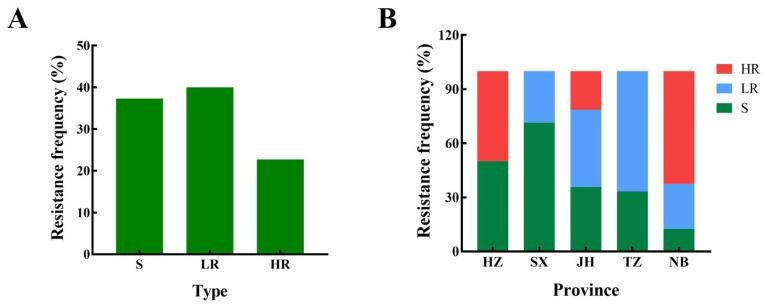
Resistance frequency of *Fusarium fujikuroi* to prochloraz in Zhejiang province (**A**) and major regions (**B**) of Zhejiang province in 2022. HZ: Hangzhou, SX: Shaoxing, JH: Jinhua, TZ: Taizhou, NB: Ningbo. R: prochloraz resistance, S: prochloraz sensitive, HR: high-level resistance, LR: low-level resistance.

**Figure 2 jof-10-00560-f002:**
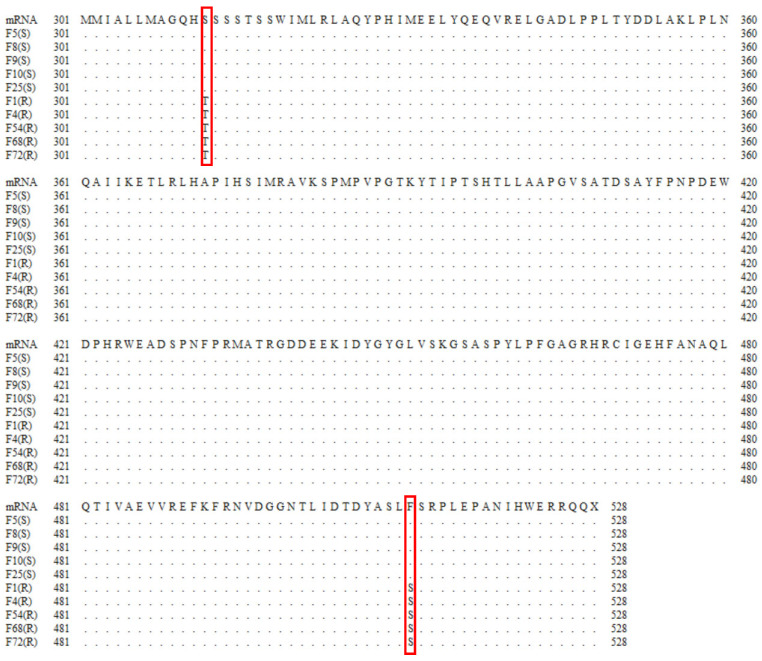
The amino acid changes of FfCYP51B were related to the resistance of *Fusarium fujikuroi* to prochloraz. Partial FfCYP51B amino acid sequences of prochloraz-sensitive (S) and -resistant (R) isolates are displayed. The red box represents the position of the amino acid substitution (codons 312 and 511).

**Figure 3 jof-10-00560-f003:**
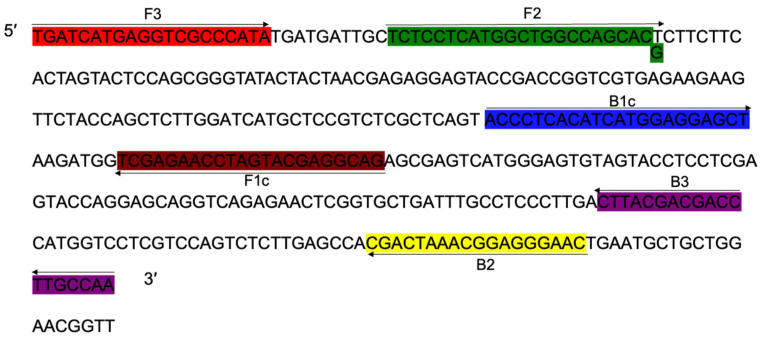
Schematic illustration of loop-mediated isothermal amplification (LAMP) primers used to detect the S312T mutant genotype of *Fusarium fujikuroi* with resistance to prochloraz.

**Figure 4 jof-10-00560-f004:**
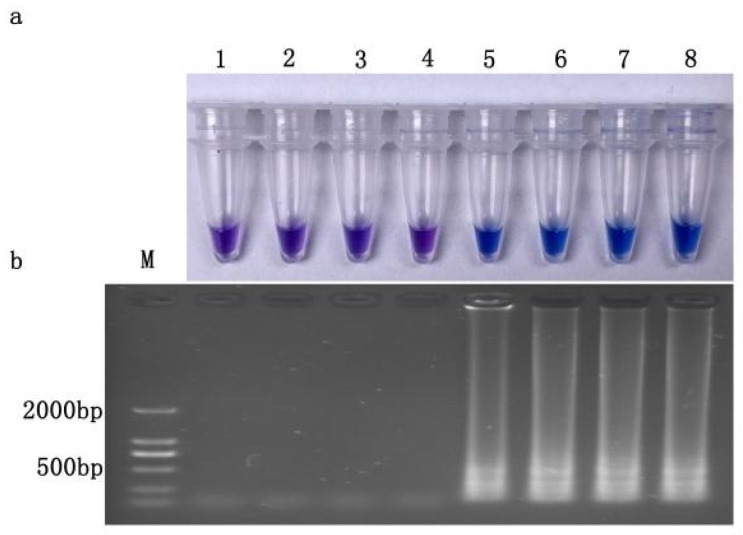
Specificity of LAMP assay for prochloraz-resistant *Fusarium fujikuroi* with S312T mutation (**a**) Hydroxynaphthol blue-visualized color change or (**b**) gel electrophoresis. M, DL2000Maker; 1, ddH_2_O; 2, JS16(prochloraz-sensitive); 3, *Ustilaginoidea virens*; 4, SX-30*^JS−16Cyp51b^*; (prochloraz-sensitive); prochloraz-resistant: 5, JH1; 6, SX23; 7, SX24; 8, JS16*^JH−1Cyp51b^.*

**Figure 5 jof-10-00560-f005:**
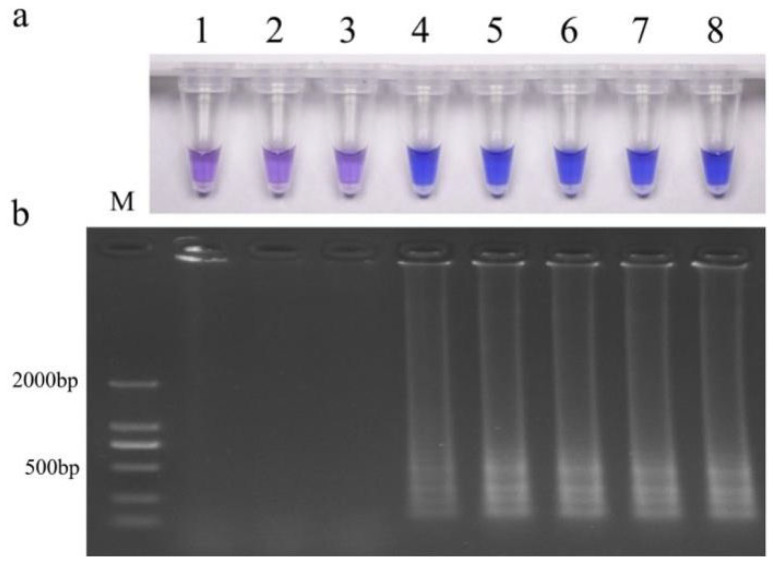
Results of LAMP detection under different DNA template concentrations. (**a**) HNB color change; (**b**) gel electrophoresis. M, DNA marker; 1, ddH_2_O; 2, 0.0001 ng/μL; 3, 0.001 ng/μL; 4, 0.01 ng/μL; 5, 0.1 ng/μL; 6, 1 ng/μL; 7, 10 ng/μL; 8, 100 ng/μL.

**Figure 6 jof-10-00560-f006:**
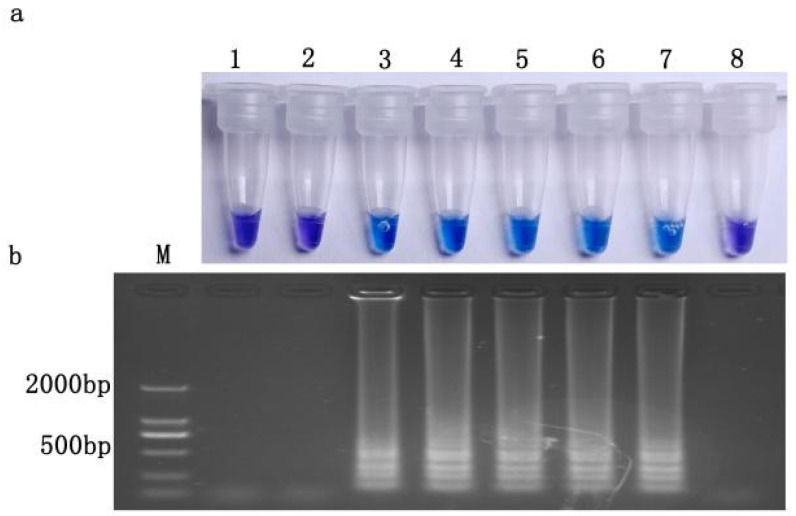
Proportion of resistant isolates detected by LAMP. (**a**) HNB color change; (**b**) gel electrophoresis. M, DL2000Maker; 1, ddH_2_O; 2, S(JS16); 3, R(JH1); 4, S:R = 10:1; 5, S:R = 20:1; 6, S:R = 100:1; 7, S:R = 200:1; 8, S:R = 400:1. S: Isolates sensitive to prochloraz; R: Resistant to prochloraz.

**Figure 7 jof-10-00560-f007:**
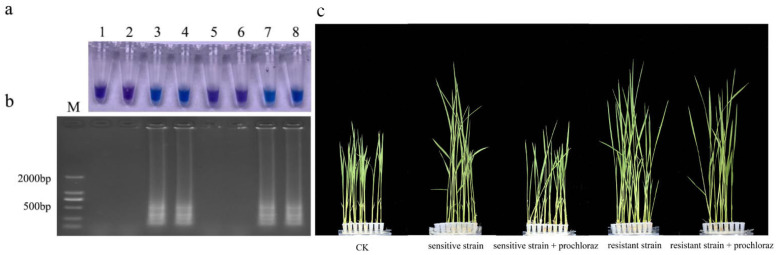
LAMP detection of prochloraz-resistant *Fusarium fujikuroi* with S312T mutation from rice seeds and seedlings. (**a**) Hydroxynaphthol blue-visualized color change; (**b**) gel electrophoresis. M, DL2000 Maker; 1–2, Sensitive isolate seeds; 3–4, Resistant isolate seeds; 5–6, Sensitive isolate seedlings + 10 μg/mL prochloraz; 7–8, Resistant isolate seedling + 10 μg/mL prochloraz; (**c**) CK; Sensitive isolate seedlings; Sensitive isolate seedlings + 10 μg/mL prochloraz; Resistant isolate seedling; Resistant isolate seedling + 10 μg/mL prochloraz.

**Table 1 jof-10-00560-t001:** LAMP reflection system.

Component (Concentration)	Final Concentration	Volume (μL) per 25 μL Reaction
Bst DNA (8 U·μL^−1^)	0.16 U·μL^−1^	0.5
10× Thermo Pol		2.5
Mg^2+^ (25 mM)	4 mM	4
dNTP (10 mM)	1 mM	2.5
F3/B3 (10 μM)	0.3 μM	0.75
F1P/B1P (10 μM)	0.8 μM	2
Betine (5 M)	0.6 M	3
HNB (3.75 mM)	150 μM	1
Sample DNA	100 ng·μL^−1^	1
ddH_2_O		5

**Table 2 jof-10-00560-t002:** Isolates used for primer design and LAMP-specific detection in this study.

Year	Fungus	Isolates	EC_50_ (μg/mL)	CODON		Resistance
				312	511	
2018	*Fusarium fujikuroi*	JS16 [20]	0.033	TCT (Ser)		S
		JH18	0.080	TCT (Ser)		S
		JH1 [20]	2.258	ACT (Thr)		R
		JH8	2.390	ACT (Thr)		R
		SX23 [20]	4.050	ACT (Thr)		R
		SX24 [20]	1.530	ACT (Thr)		R
		SX30 [20]	2.781	ACT (Thr)		R
		SX30*^JS16Cyp51b^* [20]	0.098	TCT (Ser)		S
		JS16*^JH1Cyp51b^* [20]	2.109	ACT (Thr)		R
	*Ustilaginoidea virens*	ACCC2711				
2022	*F. fujikuroi*	F1	0.242	ACT (Thr)	TCC(S)	R
		F4	1.162	ACT (Thr)	TCC(S)	R
		F54	0.852	ACT (Thr)	TCC(S)	R
		F68	0.105	ACT (Thr)	TCC(S)	R
		F72	0.232	ACT (Thr)	TCC(S)	R
		F5	0.028	TCT (Ser)	TTC(F)	S
		F8	0.037	TCT (Ser)	TTC(F)	S
		F9	0.045	TCT (Ser)	TTC(F)	S
		F10	0.016	TCT (Ser)	TTC(F)	S
		F25	0.023	TCT (Ser)	TTC(F)	S

S: Isolates sensitive to prochloraz; R: Resistant to prochloraz; JS16, JH18, JH1, JH8, SX23, SX24, SX30, F1, F4, F5, F8, F9, F10, F25, F54, F68, and F72 are wild-type isolates collected in the field. SX30*^JS16Cyp51b^* (ACT replaced by TCT) and JS16*^JH1Cyp51b^* (TCT replaced by ACT) are artificially constructed mutant isolates.

**Table 3 jof-10-00560-t003:** Biological characteristics of wild-type prochloraz-sensitive isolates and -resistant mutants of *Fusarium fujikuroi*.

Isolates	Mycelial Growth Rate (mm/d) *	Conidium Production (×10^6^/mL)	Germination Rate (%)		
			4 h	8 h	12 h
F9(S) **	(10.19 ± 0.49) a	(8.23 ± 0.56) a	(9.78 ± 1.35) a	(44.50 ± 0.94) a	(90.89 ± 0.86) a
F10(S)					
F25(S)					
F59(S)					
F1(R)	(8.27 ± 0.073) b	(6.74 ± 0.60) b	(5.22 ± 2.38) b	(30.50 ± 2.08) b	(80.75 ± 1.46) b
F4(R)					
F68(R)					
F72(R)					

* Mean ± standard error in a column followed by the same letter means not significantly different in ANOVA with LSD multiple range test at *p* < 0.05. ** S: Isolates sensitive to prochloraz; R: Resistant to prochloraz.

**Table 4 jof-10-00560-t004:** Comparison of pathogenicity to rice seedlings between wild-type prochloraz-sensitive isolates and -resistant mutants of *Fusarium fujikuroi*.

Isolates	Average Height (cm) *	Excessive Growth Rate (%)
CK	(15.30 ± 0.28) c	
F9(S) **	(19.09 ± 0.40) b	(24.74 ± 0.11) b
F10(S)		
F25(S)		
F59(S)		
F1(R)	(21.16 ± 0.38) a	(38.23 ± 0.29) a
F4(R)		
F68(R)		
F72(R)		

* Mean ± standard error in a column followed by the same letter means not significantly different in ANOVA with LSD multiple range test at *p* < 0.05. ** S: Isolates sensitive to prochloraz; R: Resistant to prochloraz.

**Table 5 jof-10-00560-t005:** Primers used in the LAMP system for detection of prochloraz-resistant *Fusarium fujikuroi* with S312T mutation.

Primer Name	Type	Sequence (5′-3′)
F3	Forward outer	TGATCATGAGGTCGCCCATA
B3	Backward outer	TTGGCAAGGTCGTCGTAAG
FIP (F1c-F2)	Forward inner primer	GACGGAGCATGATCCAAGAGCTTCATGGCTGGCCAGCACG
BIP (B1c-B2)	Backward inner primer	ACCCTCACATCATGGAGGAGCTTCAAGGGAGGCAAATCAGC

## Data Availability

Publicly available datasets were analyzed in this study. These data can be found here: https://www.ncbi.nlm.nih.gov/ (accessed on 10 December 2019).

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
