# Peer review of "Characterization, Molecular Mechanism of Prochloraz-Resistance in Fusarium fujikuroi and Development of Loop-Mediated Isothermal Amplification Rapid Detection Technique Based on the S312T Genotype of Resistances"

_jof, 2024, doi:10.3390/jof10080560_

Round 1
Reviewer 1 Report
The paper studies the Fusarium fujikuroi resistance to the imidazole fungicide Prochloraz. The paper is divided into two main parts, a study of the abundance and characterization of the resistant isolates and the development of a detection approach based on LAMP. The paper addresses the important topic of emergence of resistance to chemicals. However, the authors should have into consideration some comments to improve the paper.
The main concern is that the two parts of the paper are not connected. The ten strains that were characterized in this study (F1, F4, F5, F8, F9, F10, F25, F54, F68, and F72) were not used for the development of or tested with the LAMP approach.
The second concern is that the first part of the paper presents data that has been already published in, at least, Zhang et al. 2020 (https://doi.org/10.1002/ps.6085). An example of this is table 2, were the EC50 values are already present in the mentioned paper.
The paper seems to have been rushed for publication. The title has the name of Fusarium wrongly written. Also, the title claims that the “Molecular Mechanism of Prochloraz- Resistance” has been studied but the paper shows no data on the molecular mechanism only some SNPs used as molecular marker for resistance. Granted, the molecular mechanism has been studied in the aforementioned paper, but not here.
Overall, the paper needs to add experimental data to connect the characterization of new strains with the development of the LAMP approach. The authors should also cite in the legend of the figure/table already publish data.
The authors mention the use of “Ustilaginoidea virens” in some tables and figures (table 2, figure 4) but they do not describe why it was used.
Table 3 and 4 shows the same data for all susceptible strains, is this intended? Were different strains considered as replicates?
Some minor writing typos and expressions should be corrected, i.e. line 19 “the total resistance frequency of F. fujikuroi to prochloraz in Zhejiang Province was 62.67%, and the resistance frequency to prochloraz was higher”, the sentence seems to imply a comparation but it is not clear to what.
Author Response
Reviewer 1: The paper studies the Fusarium fujikuroi resistance to the imidazole fungicide Prochloraz. The paper is divided into two main parts, a study of the abundance and characterization of the resistant isolates and the development of a detection approach based on LAMP. The paper addresses the important topic of emergence of resistance to chemicals. However, the authors should have into consideration some comments to improve the paper. 1. The main concern is that the two parts of the paper are not connected. The ten strains that were characterized in this study (F1, F4, F5, F8, F9, F10, F25, F54, F68, and F72) were not used for the development of or tested with the LAMP approach. Response: The two parts are integrative study. First, we characterized the resistance to fungicide prochloraz and the S312T mutation in resistant isolates collected in 2022 , just as those previously described in 2018. The resistant isolates in 2022 have the same point mutation ( S312T ) as the isolates in 2018 .We performed LAMP detection based on S312T point mutation. Moreover, the following is added: For isolates of 2022, chromogenic reactions also only occurred for resistant isolates with S312T mutantion in FfCYP51B (Figure S1). 2. The second concern is that the first part of the paper presents data that has been already published in, at least, Zhang et al. 2020 (https://doi.org/10.1002/ps.6085). An example of this is table 2, were the EC50 values are already present in the mentioned paper. Response: Thank you for your comment. In previously published study, such as Zhang et al. 2020 (https://doi.org/10.1002/ps.6085),We confirmed the resistance to fungicide prochloraz was caused by the S312T mutation. In this study, we adopted some of the isolates to develop the LAMP detection based on S312T mutation. And, we have added the reference for these adopted isolates in Table2. 3. The paper seems to have been rushed for publication. The title has the name of Fusarium wrongly written. Also, the title claims that the “Molecular Mechanism of Prochloraz- Resistance” has been studied but the paper shows no data on the molecular mechanism only some SNPs used as molecular marker for resistance. Granted, the molecular mechanism has been studied in the aforementioned paper, but not here. Response: Thank you for your comment. We adjusted the title of the article. In the previous reports in China, the resistance caused by S312T mutation was found in the field and study was carried out to confirm the relationship between mutation and resistance. In this paper, the resistance caused by the double mutation of F511S and S312T was found. Overall, the paper needs to add experimental data to connect the characterization of new strains with the development of the LAMP approach. The authors should also cite in the legend of the figure/table already publish data. Response: Thank you for your comment. We have added this part of the reference in the original text and the detection for 2022 isolates. The authors mention the use of “Ustilaginoidea virens” in some tables and figures (table 2, figure 4) but they do not describe why it was used. Response: The specificity of the system was verified by using the DNA of other pathogens on a rice plant. In line 270, we changed as: Chromogenic reactions occurred in both resistant wild-type and site-specific mutant isolates, but not for sensitive isolates and U. virens, an important pathogen on rice. Table 3 and 4 shows the same data for all susceptible strains, is this intended? Were different strains considered as replicates? Response: Table 3 mainly introduces the differences in mycelial growth, Conidium production and spore germination of isolates. Table 4 mainly introduces the differences in pathogenicity. And, different strains were considered as replicates and difference between resistant and sensitive phenotypes were compared. Some minor writing typos and expressions should be corrected, i.e. line 19 “the total resistance frequency of F. fujikuroi to prochloraz in Zhejiang Province was 62.67%, and the resistance frequency to prochloraz was higher”, the sentence seems to imply a comparation but it is not clear to what. Response: Thank you for your comments. We have modified this.Reviewer 2 Report
The article is interesting because it touches on the important topic of detection of resistant isolates of a pathogen.
There are questions regarding the methodology.
Line 90-96
I think this is a repeat. The sentences should be modified to make it clear why different concentrations of prochloraz were used.
How prochloraz was added to the medium? Its solubility in water is 34.4 µg/mL, the authors use the concentration 1 and 5 µg/mL prochloraz. How were the solutions diluted?
What does «amended» mean?
The resistance of 75 F. fujikuroi isolates to prochloraz was determined using the differential dose method. Fresh plugs (5 mm in diameter) were cut from the growing edge of a mycelial culture and placed on PDA plates (90 mm in diameter) amended with 1 and 5 µg/mL prochloraz, at which concentrations the mycelial growth of the sensitive isolates was completely inhibited, while the resistant isolates continued to grow [21]. The sensitivity to prochloraz of each isolate was determined by the mycelial growth assay. Fresh plugs (5 mm in diameter) were cut from the growing edge of the colony and placed on PDA plates (90 mm in diameter) amended with 1.6, 0.4, 0.1, 0.025, and 0.00625 µg/mL prochloraz
Line 114
Did the authors try to verify how prochloraz affects the growth of conidia? To do this, it can be added to water agar.
Have you measured the optical density of solutions spectrometrically?
Author Response
Reviewer 2:
- Line 90-96
I think this is a repeat. The sentences should be modified to make it clear why different concentrations of prochloraz were used.
How prochloraz was added to the medium? Its solubility in water is 34.4 µg/mL, the authors use the concentration 1 and 5 µg/mL prochloraz. How were the solutions diluted?
What does «amended» mean?
Response: Thank you for your comment. We made changes in line 90-99: Prochloraz is dissolved in methanol as stock solution firstly and then added to the melted PDA. The word " amended " means that we adjust the concentration of prochloraz to the concentration we need. Dilution is a synonym for it. But we refer to a lot of other articles, and they all use the “amended "
- The resistance of 75 fujikuroi isolates to prochloraz was determined using the differential dose method. Fresh plugs (5 mm in diameter) were cut from the growing edge of a mycelial culture and placed on PDA plates (90 mm in diameter) amended with 1 and 5 µg/mL prochloraz, at which concentrations the mycelial growth of the sensitive isolates was completely inhibited, while the resistant isolates continued to grow [21]. The sensitivity to prochloraz of each isolate was determined by the mycelial growth assay. Fresh plugs (5 mm in diameter) were cut from the growing edge of the colony and placed on PDA plates (90 mm in diameter) amended with 1.6, 0.4, 0.1, 0.025, and 0.00625 µg/mL prochloraz
3. Line 114
Did the authors try to verify how prochloraz affects the growth of conidia? To do this, it can be added to water agar.
Have you measured the optical density of solutions spectrometrically?
Response: Thank you for your comment. We did not measure the optical density, which is good for bacteria and few of single-celled fungi such as Botrytis cinerea. We just compared the germination between resistant and sensitive isolates. NO prochloraz in the medium.
Round 2
Reviewer 2 Report
Dear colleagues, everything is fixed, I wish you good luck in further research
Comments taken into account, necessary changes made
Author Response
I read the revised manuscript and the authors responses to the reviewers’ comments and come to a decision that this manuscript still needs further revision to address the concerns from Reviewer #1, specifically the data used in Table 2. I do see that the authors have added a reference (20) in the year column and in the isolate column for JS16. However, more concerns arise after reading through the Table 2 of the published 2020 by the same group. For example: what is the relationship of this SX23 isolate in the current paper to the JH1 isolate in the Zhang et al. 2020 paper since the EC50 data is the exactly the same? I need some convincing explanation. Second, the EC50 value for SX30 has already been published in the 2020 paper, but there was no reference to the 2020 paper for this isolate. Why? A simple “oversight” will not cut it.
Response: In table 2, nine isolates of Fusarium fujikuroi were collected in 2018 (seven) or constructed mutants(two) based on isolates collected in 2018. Seven out of this nine isolates were reported in a published in the 2020 paper with information such as EC50 value. For isolate JH18, JH8 collected in 2018, no information in detail such as EC50 value was provided in the published 2020 paper. In this revised version, all the Seven were added the reference to the 2020 paper.
Also, at the end of introduction, add a couple of sentences to emphasize what new information this study adds to the existing knowledge of previous studies, such as your 2018 and the published 2020.
Response: The following was added: In this study, we want to know the evolution of resistance to prochloraz in China and clarify the newly mutation genotype associated with prochloraz-resistant in F. fujikuroi since 2018. Meanwhile, rapid detection using LAMP was further established based on the S312T point mutation in cyp51b to provide technical support for resistance detection, management and the accurate and efficient control of RBD.
Another area that I need authors to revise is the Table and Figure legends, which should contain enough information to be self-explainable, so the readers do not have to look for information in the manuscript to understand your data. For example, table 4: “Comparison of pathogenicity between wild-type prochloraz-sensitive isolates and -resistant mutants”. Need to add pathogenicity “to rice” (I assume here is rice); and need to add “sensitive isolates and -resistant mutants of “Fusarium fujikuroi”. Please double check all other figure and table legends.
Response: the Table and Figure legend were revised.